# Microbiological Findings and Associated Histopathological Lesions in Neonatal Diarrhoea Cases between 2020 and 2022 in a French Veterinary Pig Practice

**DOI:** 10.3390/vetsci10040304

**Published:** 2023-04-21

**Authors:** Gwenaël Boulbria, Charlotte Teixeira Costa, Nadia Amenna-Bernard, Sophie Labrut, Valérie Normand, Théo Nicolazo, Florian Chocteau, Céline Chevance, Justine Jeusselin, Mathieu Brissonnier, Arnaud Lebret

**Affiliations:** 1REZOOLUTION Pig Consulting Services, 56920 Noyal-Pontivy, France; 2PORC.SPECTIVE Swine Vet Practice, 56920 Noyal-Pontivy, France; 3LABOCEA, 22440 Ploufragan, France; 4CRCI^2^NA, INSERM U1307, CNRS UMR6075, Nantes University, 44007 Nantes, France

**Keywords:** piglets, neonatal diarrhoea, histopathology, *C. perfringens*, *E. hirae*, rotavirus, *E. coli*

## Abstract

**Simple Summary:**

Morbidity, mortality and loss of productivity due to enteric diseases in neonatal piglets are still major issues worldwide. Neonatal diarrhoea also represents a significant concern regarding antibiotic usage in suckling piglets. For pig veterinary practitioners, the diagnosis of neonatal diarrhoea cases is an ongoing challenge and more case reports, retrospective studies and prospective research are needed to improve knowledge. Anamnesis, clinical signs, and gross and microscopic lesions are the basics needed for a presumptive diagnosis of the pathogen(s) involved at the herd level. However, the role of some pathogens is still under discussion, which makes diagnosis all the more challenging. The aim of our retrospective study was to describe the aetiologies of neonatal diarrhoea cases in a French veterinary pig practice and to determine their associations with histological findings in the small and large intestine. Fifty one cases (48.1%) were positive for only one pathogen and 54 (50.9%) were positive for more than one pathogen. *Clostridium perfringens* type A was the most frequently detected pathogen (61.3%), followed by *Enterococcus hirae* (43.4%), rotavirus type A (38.7%) and type C (11.3%) and enterotoxigenic *Escherichia coli* (3.8%). Only *Enterococcus hirae* and rotaviruses were associated with relevant lesions in the small intestine.

**Abstract:**

This retrospective study described the aetiologies of neonatal diarrhoea cases and their associations with histological findings. A total of 106 diarrhoeic neonatal piglets were selected. Cultures, MALDI typings, PCRs and evaluation of intestinal lesions were performed. A total of 51 cases (48.1%) were positive for only one pathogen and 54 (50.9%) were positive for more than one pathogen. *Clostridium perfringens* type A was the most frequently detected pathogen (61.3%), followed by *Enterococcus hirae* (43.4%), rotavirus type A (38.7%), rotavirus type C (11.3%) and enterotoxigenic *Escherichia coli* (3.8%). Only lesions in the small intestine were correlated with detected pathogens. The detection of rotavirus was associated with an increased probability of observing villous atrophy (*p* < 0.001), crypt hyperplasia (*p* = 0.01) and leucocyte necrosis in the lamina propria (*p* = 0.05). The detection of *Clostridium perfringens* type A was associated with an increased probability of observing bacilli in close proximity to the mucosa (*p* < 0.001) and a decreased probability of observing epithelial necrosis (*p* = 0.04). Detection of *Enterococcus hirae* was associated with an increased probability of observing enteroadherent cocci (*p* < 0.001). Multivariate regression logistic models revealed that epithelial necrosis was more likely to occur in *Enterococcus hirae*-positive piglets (*p* < 0.02) and neutrophilic infiltrate was more likely to occur in *Clostridium perfringens* type A- and *Enterococcus hirae*-positive piglets (*p* = 0.04 and *p* = 0.02, respectively).

## 1. Introduction

Neonatal diarrhoea is a multifactorial condition commonly present on pig farms and leads to economic losses due to the increased morbidity and mortality of piglets. Neonatal diarrhoea also represents a significant concern regarding antibiotic usage in suckling piglets. Indeed, diarrhoea was found to be responsible for more than 21% of cases of antibiotic usage during the nursing period for piglets weaned at four weeks of age in a Danish study [1]. The prevention of neonatal diarrhoea is therefore essential for ensuring the profitability of pig herds and the reduction in antibiotic consumption. However, diagnosing the underlying cause of neonatal diarrhoea in herds is an ongoing challenge. Risk factors have been well described and include (without being exhaustive) herd size, prolificacy, management routines (e.g., maternal vaccination, hygiene procedures, cross-fostering practices, nurse sows, environmental conditions at birth, and the monitoring of farrowing), sow feeding programme (e.g., feed bacterial quality and nutritional requirements to obtain better birth weights) [2,3,4]. Beyond these risk factors, a number of infectious agents have been associated with neonatal diarrhoea in piglets. Previous studies on the prevalence of enteric pathogens involved in neonatal diarrhoea in Europe suggested that *Escherichia coli* (*E. coli*), *Clostridium perfringens* (*C. perfringens*) type C, *C. perfringens* type A, *Clostridioides difficile* (*C. difficile*), *Enterococcus hirae* (*E. hirae*), coronaviruses, rotaviruses and *Cystoisospora suis* might be of significance [5]. However, the detection of a pathogen alone is not sufficient for elucidating the aetiology behind neonatal diarrhoea outbreaks and is insufficient for helping practitioners design preventative programmes. Relevant associated lesions must be considered simultaneously with enteric pathogen detection to achieve a correct diagnosis of the agent involved. Studies describing the observation of histopathological lesions associated with enteric pathogen detection in the context of neonatal diarrhoea in the field via light microscopy are scarce [6,7,8], either because they focus on a single pathogen [9,10] or because they are based on the detection of agents in faeces without histopathological examination [11,12,13,14,15]. The aim of this retrospective study was to describe the frequency of enteric pathogens contributing to neonatal diarrhoea in a French veterinary pig practice and to investigate the association between enteric pathogen detection and intestinal histopathological lesion observations. We limited our detection of agents to toxigenic *E. coli* (carrying genes for LT, Sta and STb), *C. perfringens*, *E. hirae*, rotavirus type A (RVA) and rotavirus type C (RVC).

## 2. Materials and Methods

### 2.1. Herd and Animals

In this study, all piglets (n = 106) under one week of age selected between January 2020 and October 2022 for neonatal diarrhoea diagnosis in a French veterinary pig practice were retrospectively included in this study. The piglets selected by veterinarians were from farms in which diarrhoea was observed in more than 20% of litters within a batch for at least two consecutive batches, and in some herds despite receiving maternal vaccinations. They showed watery to creamy diarrhoea starting in the previous 24 h, but no other clinical signs (e.g., weight loss, arthritis or omphalitis). Neither the piglet nor its dam were treated with antibiotics from birth to selection. These piglets originated from 38 French farms, mostly in the north-west of the country, where the high-density pig breeding areas are located. In each herd, two to four diarrhoeic piglets under one week of age were selected from different litters. The piglets were transported live to the closest diagnostic laboratory for euthanasia, post-mortem examination and sampling (Laboratory 1: Labocéa, Ploufragan, France; Laboratory 2: Labofarm, Loudéac, France). Both laboratories are ISO 17025-certified and officially accredited by Cofrac^®^ (Paris, France) for animal health analyses (licences n° 1-7015 for Laboratory 1 and n° 1-7231 for Laboratory 2).

### 2.2. Necropsy Procedures

In Laboratory 1, piglets were euthanised using a two-stage (head-only followed by head-to-heart) mobile electric stunner (Euthazen^®^, Cimac Elevage, Ploufragan, France). In Laboratory 2, the piglets were first sedated with an intramuscular injection of 6 mg/kg of tiletamine and zolazepam(Zoletil 50^®^, Virbac Animal Health, Carros, France) and euthanised with an intracardiac injection of 1.3 mg/kg of tetracaine, 60 mg/kg of embutramide and 8.1 mg/kg of mebezonium (T61^®^, MSD Animal Health, Beaucouzé, France). In both laboratories, necropsies were performed immediately after death. Tissue specimens were collected from the proximal duodenum, proximal jejunum, distal jejunum, ileum and colon. Specimens were flushed and then fixed with 10% neutral-buffered formalin for 24 h.

### 2.3. Microscopic Observations

Formalin-fixed samples were dehydrated, embedded in paraffin blocks, sectioned at 4 µm, and stained with haematoxylin, eosin and saffron by Labocéa. Intestinal sections were evaluated by light microscopy by investigators blinded to the results of the bacteriological and virological investigations. They were observed using Leica DM 2000 and 2500 (Leica Microsystems, Wetzlar, Germany) at different magnifications (×25 to ×630).

### 2.4. Bacteriological Investigations

Bacteriological samples were collected promptly after euthanasia from the proximal duodenum, the distal jejunum, the distal ileum and the proximal colon using a 1 µL inoculation loop to collect intestinal content. Specimens were cultured within 30 min of sampling. For *E. coli*, specimens were cultured aerobically on 5% sheep’s blood agar plates and Drigalski lactose agar plates at 37 °C. The plates were examined for bacterial growth after 24 h of incubation. For *Clostridium* spp., specimens were cultured anaerobically on 5% sheep’s blood agar plates and nalidixic acid agar plates at 37 °C and incubated for 24 h. *Enterococci* were cultured aerobically at 37 °C on nalidixic acid agar plates. All bacterial species were identified using MALDI–TOF (matrix-assisted laser desorption ionization–time of flight) mass spectrometry (Brucker Daltronics, Bremen, Germany). The growth of *C. perfringens* was quantitatively assessed. A piglet was considered positive for *C. perfringens* if at least ten colonies in either the jejunum or ileum grew on the plate.

### 2.5. E. coli Virulence and Adherence Factors Detection

When β-haemolytic *E. coli* colonies were present, two colonies per piglet and per section were selected and pooled for genotyping. Otherwise, two typical non-haemolytic colonies were chosen. In Laboratory 1, *E. coli* colonies were tested for the presence of genes coding heat-labile enterotoxin (LT) (target gene *eltB*), heat-stable toxin a (STa) (target gene *estA*) and b (STb) (target gene *estB*), and for genes coding fimbrial adhesins F4 (target gene *faeG*), F5 (target gene *fanC*), F6 (target gene *fasA*), F18 (target gene *fedA*) and F41 (target gene *fim41A*) using an in-house multiplex real-time PCR. Samples with a cycle threshold (Ct) of <30 and a curve showing a specific exponential shape were considered positive. In Laboratory 2, *E. coli* colonies were tested for the presence of genes coding LT, STa and STb. LT (target gene *eltB*), STa (target gene *estA*) and STb (target gene *estB*) were detected by an in-house PCR. *E. coli* colonies were also tested for fimbrial adhesins (F4, F5, F6, F18 and F41) using agglutination tests with monovalent F-antisera. In our study, a piglet was considered enterotoxigenic *E. coli*-positive if at least one of the two tested isolates was positive for one or more virulence factors and fimbrial adhesins.

### 2.6. C. perfringens Genotyping

Four *C. perfringens* colonies from each *C. perfringens*-positive piglet were pooled and typing was performed using multiplex in-house PCRs for the detection of genes coding the α toxin (Cpα) (target gene *cpa*), the β toxin (Cpβ) (target gene *cpb*) and the β2 toxin (Cpβ2) (target gene *cpb2*). *C. perfringens* isolates were identified as type C strains if they possessed the Cpβ genes, and as type A strains if they possessed only the Cpα genes and possibly the Cpβ2 genes.

### 2.7. Rotaviruses Detection

All samples were analysed in Laboratory 2 for rotavirus detection. Two single-plex real-time PCR assays were used following the manufacturer’s recommendations: the Kylt real-time PCR for porcine rotavirus type A kit and the Kylt real-time PCR for porcine rotavirus type C kit (AniCon Labor GmbH, SAN group, Hoeltinghausen, Germany). Samples with Ct < 42 and a curve showing a specific exponential shape were considered positive.

### 2.8. Statistical Analyses

Data management, and the determination of the descriptive frequencies of enteric pathogen detection in the cases were performed using Excel v.22.10 (Microsoft Corporation, Redmond, WA, USA). Descriptive analyses and statistics were calculated using R version 4.2.2 (R Core Team, R Foundation for Statistical Computing. R: a language and environment for statistical computing, https://www.r-project.org/ (accessed on 4 December 2022). For each analysis, the upper limit for the statistically significant effect was set at *p* ≤ 0.05, with 0.05 < *p* ≤ 0.10 considered a tendency. For all analyses, all variables were collapsed into binary variables as present or absent. The pathological lesions were considered regardless of their severity. The proportion of cases in which a lesion was found was calculated for each pathogen. The association of pathogen detection with that of histopathological lesions was assessed by Fisher’s exact tests (FET).

Multivariate logistic regression analyses on the associations between enteric pathogen detection and microscopic lesion detection were performed using the logit link to estimate the odds ratio (OR) along with their respective 95% confidence intervals (CI). These models were built by gradually removing the less significant variables until a set of variables remained that produced the highest model accuracy, according to the lowest Akaike information criterion value obtained.

## 3. Results

### 3.1. General Information on Diarrhoea Cases

The piglets ranged in age from <1 day to 7 days (median age 4.25 days). Gender was recorded for 62 piglets; 32 were female and 30 were male. Finally, weight at necropsy was recorded for 72 cases and was on average 1.42 kg (±0.33 kg). Eighty-four percent of the selected farms implemented maternal vaccination against neonatal diarrhoea; 82% implemented vaccination against enterotoxigenic *E. coli*, 75% implemented vaccination against *C. perfringens* type C, 21% implemented vaccination against *C. perfringens* type A and 3% implemented vaccination against rotavirus type A using the full dose of bovine vaccines.

### 3.2. Rate of Detection of Enteric Pathogens in Neonatal Diarrhoea Cases

All piglets but one were positive for at least one of the examined enteric pathogens. A total of 51 out of 106 (48.1%) were positive for only one of these pathogens, and 54 out of 106 (51%) were positive for more than one pathogen. All farms were positive for at least one enteric pathogen as found in the piglets sampled for neonatal diarrhoea diagnosis. *C. perfringens* was the most frequently detected pathogen, both in piglets (61.3%) and on farms (73.7%) (Table 1). Cpα and Cpβ2 toxin genes were detected in all strains. *E. hirae* was isolated in 43.4% of the piglets and 57.9% of the farms. *E. coli* strains were isolated from all piglets submitted to the laboratory but the percentage of cases in which enterotoxigenic *E. coli* was isolated was low (3.8% of the piglets and 7.9% of the farms). The four isolates carried the F5 adhesin and the Sta gene. Regarding rotaviruses, RVA was detected in 38.7% of the piglets submitted and 50% of the farms, and RVC was detected in 11.3% of the piglets and 10.5% of the farms.

### 3.3. Simultaneous Detection of Enteric Pathogens at Piglet Level

*C. perfringens* type A was isolated simultaneously with at least another pathogen in 42 cases (64.6% of *C. perfringens* type A-positive cases) (Table 2). The most frequent association detected was that between RVA and *C. perfringens* type A (17.9% of piglets submitted). RVC was never detected alone; it was detected mainly with *C. perfringens* type A (five piglets), with RVA (two piglets), with both *C. perfringens* type A and *E. hirae* (two piglets), with both RVA and *C. perfringens* type A (one piglet) and with *E. hirae* (one piglet).

### 3.4. Associated Histopathological Lesions

Some microscopic lesions are illustrated in Figure 1. The five most frequently observed lesions in the small intestine were those associated with inflammatory infiltrates in the lamina propria (99 piglets), bacilli in close proximity to the mucosal surface (85 piglets), an abundance of neutrophils in the inflammatory infiltrate (84 piglets), the shortening of villi (74 piglets), and the necrosis of epithelial cells (64 piglets). The three most frequently observed lesions in the colon were those associated with mesocolonic oedema (68 piglets), inflammatory infiltrates (48 piglets) and oedema (22 piglets) in the lamina propria. Enterotoxigenic *E. coli* was excluded from the following analyses because of the small number of positive cases.

#### 3.4.1. Association between Single Histopathological Lesions and Single Enteric Pathogen Presence

In this analysis, the detection of enteric pathogens was carried out by determining either their presence or absence, without taking into account the possible simultaneous detection of different pathogens. The distribution of pathogen detection and of each single lesion in the cases (as a proportion of the total number of cases in which the pathogen was detected) are depicted in Figure 2.

As reported in Table 3, the detection of rotavirus was associated with the following lesions: those associated with villous atrophy (VA) (FET: *p* < 0.001; OR = 8.38; 95% CI, 2.90–24.24), crypt hyperplasia (CH) (FET: *p* = 0.01; OR = 5.58; 95% CI, 1.47, 21.14), enteroadherent cocci (EC) (FET: *p* < 0.001; OR = 0.16; 95% CI, 0.06–0.43), oedema in the lamina propria (OED) (FET: *p* = 0.04; OR = 0.43; 95% CI, 0.20–0.94), and leucocyte necrosis in the lamina propria (LPN) (FET: *p* = 0.05; OR = 2.30; 95% CI, 1.06–5.02). The detection of rotavirus made the observation of villous atrophy (Figure 1A), crypt hyperplasia and leucocyte necrosis (Figure 1B) in the lamina propria more probable and made the observation of enteroadherent cocci (Figure 1C) and oedema in the lamina propria (Figure 1D) less probable.

The detection of *C. perfringens* type A was associated with that of the lesions associated with epithelial necrosis (EN) (FET: *p* = 0.04; OR = 0.40; 95% CI, 0.17–0.94) and enteroadherent cocci (FET: *p* < 0.001; OR = 0.19; 95% CI, 0.08–0.47), and the observation of bacilli in close proximity to the mucosa (FET: *p* < 0.001; OR = 5.67; 95% CI, 1.98–16.26). The detection of *C. perfringens* type A made the observation of bacilli in close proximity to the mucosa (Figure 1E) more probable and made the observation of epithelial necrosis (Figure 1F) and enteroadherent cocciless probable.

Finally, the detection of *E. hirae* was associated with the lesions associated with enteroadherent cocci (FET: *p* < 0.001; OR = 54.38; 95% CI, 11.72–252.28) and the observation of bacilli in close proximity to the mucosa (FET: *p* = 0.02; OR = 0.30; 95% CI, 0.11–0.83). The detection of *E. hirae* made the observation of enteroadherent cocci more probable and the observation of bacilli in close proximity to the mucosa less probable.

No significant associations (using FET) between pathogen detection and histopathologic lesions in the colon were detected.

#### 3.4.2. Association between Intestinal Histopathological Lesions and Enteric Pathogen Detection (Alone or with Simultaneous Detection of Others) Using Multivariate Analyses

Multivariate regression logistic models were applied, using lesions (of villous atrophy, vascular congestion, epithelial necrosis and neutrophilic infiltrate) as dependant variables and pathogens as covariates. The following results are valid whether the pathogen was detected alone or with others. For villous atrophy, the best model included all cases in which rotaviruses were detected. Indeed, the presence of villous atrophy was more likely to occur in diarrhoeic pigs in which rotavirus was detected (OR = 8.70; 95% CI 2.92–30.52, *p* < 0.001). For vascular congestion in the lamina propria (Figure 1G), the best model included all cases in which rotaviruses were detected. Indeed, the presence of vascular congestion in the lamina propria was more likely to be observed in diarrhoeic piglets in which rotaviruses were detected (*p* = 0.07). For epithelial necrosis, the best model included all cases in which *E. hirae* was detected. The presence of epithelial necrosis was more likely to occur in diarrhoeic piglets in which *E. hirae* was detected (OR = 7.50, 95% CI 1.52–42.57, *p* < 0.02). Finally, for neutrophilic infiltrates (Figure 1H), the best model included all cases in which *C. perfringens* type A and *E. hirae* were detected. The presence of a neutrophilic infiltrate was more likely to be observed in diarrhoeic piglets in which *C. perfringens* type A (OR = 3.37, 95% CI 0.86,13.52, *p* = 0.04) was detected. Similarly, the presence of a neutrophilic infiltrate was more likely to be observed in diarrhoeic piglets in which *E. hirae* (OR = 8.62; 95% CI 1.54–58.79, *p* = 0.02) was detected.

## 4. Discussion

This retrospective and descriptive study was designed to aid the better understanding and more accurate diagnosis of neonatal diarrhoea by veterinary pig practitioners. The diagnosis of neonatal diarrhoea is one of the most challenging ones in daily pig practice. There are two issues: first, pig veterinarians might detect more than one enteric pathogen and have to prioritize advice for the prevention of one of these pathogens; secondly, the current understanding of some pathogens such as *C. perfringens* type A, *E. hirae*, RVC and *C. difficile* as significant pathogens is difficult to substantiate, although vaccines are sometimes marketed for some of them. In addition to an attentive clinical and epidemiological investigation and pathogen detection (by culture and/or PCR), a definite diagnosis will be achieved following a histological examination of specimens and the demonstration of profuse colonization by bacteria in close proximity to or adhering to the mucosa [5]. However, as reported in this study and in many previous studies, practitioners often detect more than one suspected enteric pathogen and no lesion is pathognomonic for any of these pathogens. The final conclusions and preventive recommendations provided by the practitioner are often a presumptive diagnosis and can be influenced by economic constraints. There is still a paucity of publications for strong-evidence-based neonatal diarrhoea diagnosis.

The present study reports data on the rate of detection of the main pathogens associated with neonatal diarrhoea in a French veterinary pig practice. To reach a diagnosis as accurately as possible, the procedure is standardised in the practice; piglets under one week of age must be submitted alive to the laboratory in the presumed acute stage of the disease and without previous treatment of the piglet or the dam. Under these conditions, submitted cases are the most likely to provide useful information for a proper diagnosis combining a bacteriological culture of intestinal sections, rotavirus detection by PCR and histopathological examination.

Enterotoxigenic *E. coli* and *C. perfringens* type C are well-known major causes of neonatal diarrhoea [5] but their clinical importance seems to have decreased as previously reported [9,11,13,14,16] and as confirmed in this study. Reasons might include the implementation of vaccination programmes against both pathogens (82% and 75% of the piglet’s dams were vaccinated in this study) and the improvement of hygiene procedures in modern pig production [5]. In our study, enteropathogenic *E. coli* was not investigated. Indeed, enteropathogenic *E. coli* is usually related to post-weaning diarrhoea [17]. Noteworthily, enteropathogenic *E. coli* isolates from neonatal diarrhoea cases have been previously reported [13,14,18,19].

The *C. perfringens* strains cultured in this study were *C. perfringens* type A isolates and both Cpα and Cpβ2 toxin genes in all the tested isolates. The role of *C. perfringens* type A as a primary cause of neonatal diarrhoea is controversial [5,20]. *C. perfringens* type A is commonly found in healthy piglets [9,14,16,21]. Since *C. perfringens* type A is a common bacterium of gut microbiota, its detection cannot be interpreted unambiguously because to date it is impossible to distinguish commensal strains from pathogenic strains [14]. *C. perfringens* type A was detected in 60.7% of diarrhoeic cases by Vidal et al., (2019) which is close to the rate of detection in our study and in a larger proportion (89.9% of submissions) by Mesonero-Escuredo et al. (2018) in Spain. In this last report, *C. perfringens* type A was detected in 43.1% of cases in association with other pathogens [13] compared to 77.8% of cases in our study. This high rate of detection in association with other enteric pathogens in our study might suggest that *C. perfringens* type A was rarely the primary agent of neonatal diarrhoea. In our study, a piglet was considered positive for *C. perfringens* type A if at least ten colonies in either the jejunum or ileum grew on the plate, but it is still difficult to know if it acted as primary pathogen or if its overgrowth was triggered by other infectious or non-infectious factors. Moreover, no evidence for the adhesion of the bacteria to epithelial cells or associated microscopical lesions has been described to date [5,6]. In our study, the detection of *C. perfringens* type A increased the probability of observing bacilli in close proximity to the mucosa and decreased the probability of observing epithelial necrosis. Degenerative and necrotic changes in the intestinal mucosa have been described as being suggestive of clostridiosis in pigs [20,22]. However, the presence of bacilli in close proximity to the mucosa did not correlate with that of simultaneous histological lesions in one previous study, suggesting that the localization of *C. perfringens* in the intestinal mucosa is not linked to its pathogenicity [6], which could be confirmed in our study. It is believed that *C. perfringens*’ toxins play a significant role in its pathogenicity but it still needs further investigation.

The practice of diagnosing of rotavirus-associated neonatal diarrhoea based on the mere presence of the virus (using PCR assays or ELISA quick tests) without the observation of any histopathological changes is debatable. RVA was detected by PCR in 38.7% of cases in our study, which is close to the rate of detection in Spain (43.1–51.6%) [13,14] and in Germany (35%) [12], but higher than the rate of detection in Denmark (9–25%) [11,23] and in Sweden [24]. RVA was the only pathogen that could be associated with neonatal diarrhoea in two recent case–control studies [11,14]. Other publications reported the relationship between RVA infection and neonatal diarrhoea cases, with or without coinfections [13,25], and sometimes combined these with management discrepancies [25]. RVC was detected by PCR in 11.3% of cases in our study, which is lower than the rate of detection in Spain (33.6%) [14]. The role of RVC in neonatal diarrhoea is controversial since a previous study reported that the prevalence of RVC was similar bewteen diarrhoeic and healthy piglets [6]. Moreover, as in our study, Vidal et al. (2019) found that RVC-positive cases were also positive for another enteric pathogen (RVA in their study). However, a recent study in Australia reported histopathological changes suggestive of rotavirus infection together with the detection of only RVC in six neonatal diarrhoea cases [26], and another in the USA also suggested a pathological role for RVC [27].Therefore, the simultaneous detection of RVC with another enteric pathogen may not have excluded its pathogenic role in our study. Lesions from RVA, RVB and RVC are similar [27,28,29,30]. Villous atrophy [27,28,29], elongated intestinal crypts [27,28], submucosal oedema [27], lymphohistiocytic enteritis [27] and epithelial vacuolar degeneration [27] were described as rotavirus infection-associated lesions. In our study, the detection of rotaviruses increased the probability of observing villous atrophy, crypt hyperplasia and leucocyte necrosis in the lamina propria and decreased the probability of observing oedema in the lamina propria of the small intestine. Moreover, multivariate logistic regression models revealed that the presence of villous atrophy was more likely to occur and the presence of vascular congestion in the lamina propria was more likely to occur in diarrhoeic piglets in which rotaviruses were detected (*p* = 0.07).

*Enterococcus* sp. is suggested to be of increasing importance in neonatal diarrhoea, although its pathogenesis remains unknown. *E. hirae* was the second most frequently isolated pathogen in our study, being detected in 43.4% of cases, which is consistent with the finding of a Danish report (44%) [11]. *E. hirae* is one of the three species of particular importance (*E. villorum*, *E. durans* and *E. hirae*) that have been associated with neonatal diarrhoea in piglets. Some studies have reported evidence of its role as a primary agent of neonatal diarrhoea [6,9,10,31,32]. Histopathological changes described in *E. hirae* infection cases were the presence of enteroadherent Gram-positive cocci in abondance associated with villous epithelial damage and atrophy [10]. In our study, the detection of *E. hirae* made the observation of enteroadherent cocci more probable. This lesion is noteworthy since there was a significant positive correlation between the adherence of *Enterococcus* spp. and neonatal diarrhoea occurrence [6]. Moreover, multivariate logistic regression models revealed that the presence of epithelial necrosis and of a neutrophilic infiltrate in the lamina propria of the small intestine was more likely to occur in the diarrhoeic piglet in which *E. hirae* was detected. The mechanisms by which *Enterococcus* spp. cause diarrhoea remain unclear and need further investigation.

Several studies have suggested that PRRSV contributes to the development of neonatal diarrhoea, particularly those that demonstrate PRRSV-infected macrophages in the intestinal lamina propria [8,33]. In our study, the PRRSV status of farms was not investigated but no clinical evidence of PRRSV instability was noticed by veterinary practitioners. Finally, *C. difficile* was not investigated in our study. Indeed, during the last ten years in France, *C. difficile*-associated microscopic lesions have been rarely observed by pig pathologists and the pathogen is rarely cultured. Therefore, this pathogen is probably an unlikely cause of neonatal diarrhoea in France. The mere detection of toxins (which are also detected in healthy piglets [16]) and the observation of mesocolonic oedema (which is a very common finding in our cases) should not be interpreted as the cause of *C. difficile*-associated neonatal diarrhoea without the observation of typical volcano lesions in the colon [5].

## 5. Conclusions

The diagnosis of the underlying cause of neonatal diarrhoea in pig herds is an ongoing challenge. In addition to clinical and epidemiological findings, relevant associated lesions must be considered simultaneously with enteric pathogen detection to identify pathogens involved in the disease. Indeed, our study emphasizes the challenge of neonatal diarrhoea diagnosis for swine vet practitioners from laboratory reports with several enteric pathogens being detected in the same case and the diversity and variability of lesions in histopathological reports being observed. However, this study reports histopathological findings that might be considered together with bacterial culture and PCR to carry out diagnoses in a field context. Moreover, this study provides evidence of the role of rotavirus and *Enterococcus hirae* as enteric pathogens involved in neonatal diarrhoea. Finally, the pathogenic role of *Clostridium perfringens* type A is still debatable and its detection in our study was not significantly correlated with the observation of major microscopic lesions.

## Figures and Tables

**Figure 1 vetsci-10-00304-f001:**
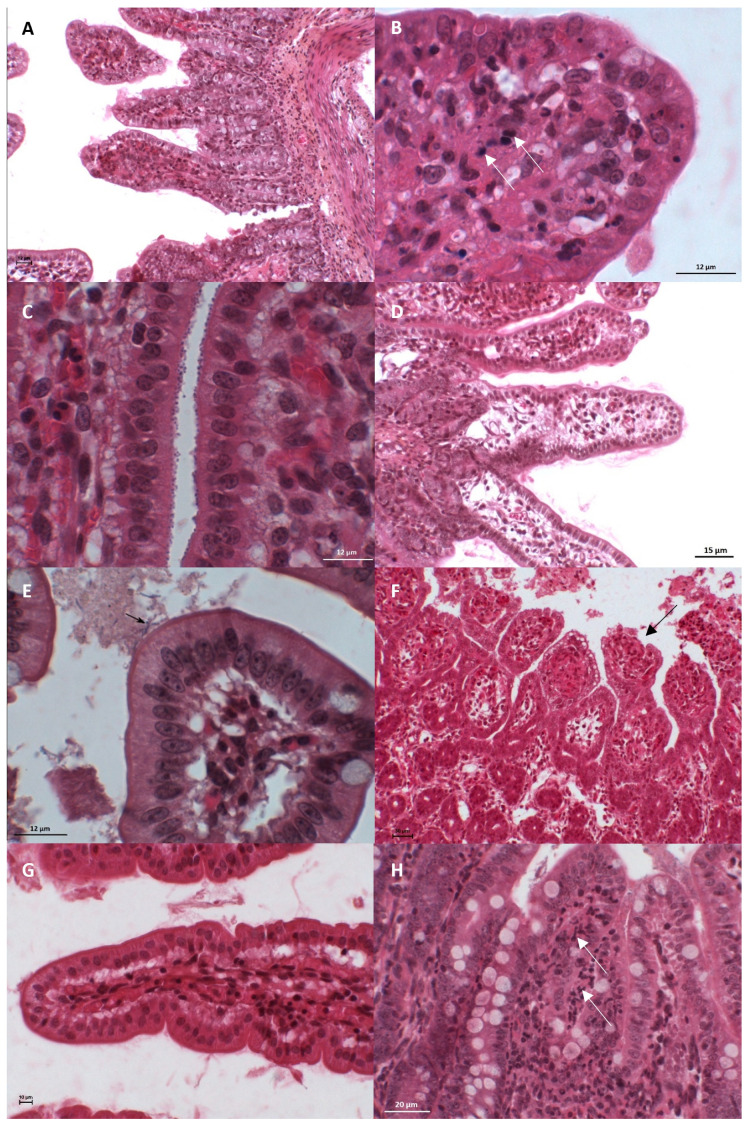
Jejunal histological sections stained with haematoxylin, eosin and saffron in piglets affected by neonatal diarrhoea illustrating the main lesions reported in this study. (**A**) Villous atrophy of a rotavirus type A-infected piglet. Scale bar: 12 µm. (**B**) Leucocyte necrosis (arrows) of a rotavirus type A-infected piglet. Scale bar: 12 µm. (**C**) Colonisation by enteroadherent cocci of an *Enterococcus hirae*-infected piglet. Scale bar: 12 µm. (**D**) Oedema in the lamina propria of an *Enterococcus hirae*-infected piglet. Scale bar: 15 µm. (**E**) Observation of bacilli (arrow) in close proximity to the epithelium of a piglet positive for rotavirus type A and *Clostridium perfringens* type A. Scale bar: 12 µm. (**F**) Epithelial necrosis (cell degeneration and epithelial ulceration) (arrow) of an *Enterococcus hirae*-infected piglet. Scale bar: 30 µm. (**G**) Vascular congestion in the lamina propria of a rotavirus type A-infected piglet. Scale bar: 10 µm. (**H**) Neutrophil infiltration (arrows) in the lamina propria of a piglet positive for rotavirus type A and *Clostridium perfringens* type A. Scale bar: 20 µm.

**Figure 2 vetsci-10-00304-f002:**
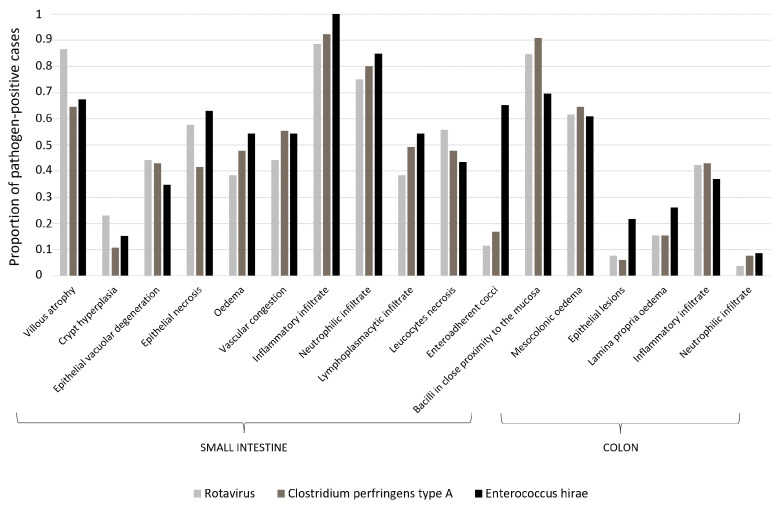
Proportion of pathogen-positive cases in which a lesion was observed.

**Table 1 vetsci-10-00304-t001:** Rate of detection of bacterial agents and rotaviruses in samples from diarrhoeic piglets.

Pathogen	Piglets (n = 106)		Farms (n = 38)	
	Number	%	Number	%
Bacterial agents isolated by culture				
*Clostridium perfringens* type A	65	61.3	28	73.7
*Enterococcus hirae*	46	43.4	22	57.9
Enterotoxigenic *Escherichia coli*	4	3.8	3	7.9
Rotaviruses detected by PCR				
Rotavirus type A	41	38.7	19	50.0
Rotavirus type C	11	11.3	4	10.5

**Table 2 vetsci-10-00304-t002:** Simultaneous detection of different pathogens in diarrhoeic piglets.

Simultaneous Detection of Pathogens	Piglets (n = 106)	
	Number	%
Rotavirus type A (RVA) + *Clostridium perfringens* (*C. perfringens*) type A	19	17.9
*C. perfringens* type A + *Enterococcus hirae* (*E. hirae*)	9	8.5
RVA + *C. perfringens* type A + *E. hirae*	7	6.6
Rotavirus type C (RVC) + *C. perfringens* type A	5	4.7
RVA + *E. hirae*	4	3.8
*C. perfringens* type A + *Escherichia coli* (*E. coli*)	3	2.8
RVA + RVC	2	1.9
RVC + *C. perfringens* type A + *E. hirae*	2	1.9
*C. perfringens* type A + *E. hirae* + *E. coli*	1	0.9
RVA + RVC + *C. perfringens* type A	1	0.9
RVC + *E. hirae*	1	0.9

**Table 3 vetsci-10-00304-t003:** Number of pathogens isolated and number of associated histopathological lesions in the small intestine.

		Epithelium							Lamina Propria					
Pathogens	Isolated	VA	CH	VD	EN	EC	BCP		OED	VC	IFI	NEI	LYI	LN
Rotavirus	52	**45 *^I^**	**12 *^I^**	23	30	**6 *^D^**	44		**20 *^D^**	23	46	39	20	**29 *^I^**
*C. perfringens* type A	65	42	7	28	**34 *^D^**	**11 *^D^**	**59 *^I^**		31	36	60	52	32	31
*E. hirae*	46	31	7	16	29	**30 *^I^**	**32 *^D^**		25	25	46	39	25	20

Association between detection of a pathogen (yes/no) and that of pathological lesions (yes/no) calculated by Fisher’s exact test. * Statistically significant association. I = presence of the pathogen is associated with an increased probability of detecting a lesion. D = the presence of the pathogen is associated with a decreased probability of detecting a lesion. VA: villous atrophy; CH: crypt hyperplasia; VD: vacuolar degeneration; EN: epithelial necrosis; EC: enteroadherent cocci; BCP: bacilli in close proximity to the mucosa; OED: oedema; VC: vascular congestion; IFI: inflammatory infiltrate; NEI: neutrophilic infiltrate; LYI: lymphoplasmacytic infiltrate; LN: leucocyte necrosis in the lamina propria.

## Data Availability

The data that support the findings of this study are available from the corresponding author upon reasonable request.

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
