# Peer review of "Microbiological Findings and Associated Histopathological Lesions in Neonatal Diarrhoea Cases between 2020 and 2022 in a French Veterinary Pig Practice"

_vetsci, 2023, doi:10.3390/vetsci10040304_

Round 1
Reviewer 1 Report
The present paper describes the aetiologies of neonatal diarrhoea cases and their associations with histological findings. The authors selected 106 piglets under one week of age from 38 French farms experiencing diarrhoea in more than 20% of litters and then perform Cultures, MALDI-typings, PCRs and evaluation of intestinal lesions. The agents detected included toxigenic E. coli (carrying genes for LT, Sta and STb), C. perfringens, E. hirae, rotavirus type A (RVA) and rotavirus type C (RVC). The results showed that 48,1% of cases were positive for only one pathogen and 50,9% were positive for more that one pathogen. The presence of rotavirus was positively correlated to villous atrophy, crypt hyperplasia and leucocytes necrosis in the lamina propria, detection of C. perfringens risulted positively associated with an increased probability of observing bacilli in close proximity to the mucosa and the presence of Enterococcus hirae was associated with an increased probability of observing enteroadherent cocci.
The work is well written, methodologically accurate and the results are properly disscussed. As is well known, some of the selected pathogens are also commonly found in healthy piglets. This makes it particularly difficult to study the correlations of such agents with neonatal diarrhea. In any case, the authors are aware of this limitation that is well discussed. In addition, as the authors report the percentage of herds that implemented maternal vaccination (179-182), it would be interesting to analyse the results according to the origin of the piglets (non maternal-vaccinated farms versus maternal-vaccinated farms). Finally, as a number of infectious agents have been associated with neonatal diarrhoea in piglets, it would be useful to extend the spectrum of pathogen considered to obtein a more complete evaluation of the problem.
Author Response
Dear reviewer,
many thanks for your attentive review and your precious comments for future analyses. We are still recording data from cases in our practice and in few years, we hope to have enough cases to do such comparisons.
Reviewer 2 Report
The manuscript provides interesting information regarding the accurate diagnosis of neonatal diarrhoea, a challenging diagnostic in swine farms nowadays. Although this diagnosis is usually based on microbiological detection of different microorganisms by culture or PCR often it does not allow for an etiological diagnosis since many of them are opportunistic or infection involves virulence factors not well established. In this research, association of microbiological detection with histological lesions was investigated and provided relevant results. Some questions that arise from the research conducted include:
- - The relevance of histological monitoring of lesions in the diagnosis of neonatal diarrhoea is demonstrated. Do the authors considered that quantitative detection using q-PCR would also provide relevant information? I think this point is interesting to discuss. Also, immunohistochemical detection of pathogens should be considered because it would be the gold standard method to evaluate the pathological role of different microorganisms. Can this method be applied in the histological samples for further research?
- - The detection of etiological agents was limited to ETEC, C. perfringens, E. hirae, RVA and RVC. It is unclear why Clostridiodes difficile and enteropathogenic E. coli (EPEC) were not include in the research. The authors should clearly justify the selection of pathogens included in the research.
- - In the discussion section (lines 383-389), it is mentioned the pathogenic role of C. difficile (not Clostridium but Clostridiodies) is debatable and unclear. However, it is generally considered that C. difficile is a relevant pathogen for young piglets (usually recognized as a more clearly pathogenic as compared to C. perfringens type A). Also, it is mentioned that C. difficile was not included in the research since there have been a significant reduction of antibiotic use in French pig herds. However, there is to my knowledge no relationship between the use of antibiotics and C. difficile detection in pigs has been demonstrated (in contrast to C. difficile infections in humans).
- - It is mentioned that “The role of RV as primary pathogens in neonatal diarrhoea is also reported as debatable” (lines 337-338). However, many reports have described their involvement in diarrhoea outbreaks in young animals, including both RVA and RVC. It is also mentioned that “The role of RVC in neonatal diarrhoea is controversial since, as in our study, Vidal et al found that RVC positive-cases were also positive for another enteric pathogen (RVA in their study)”. In my personal opinion, the fact that two different pathogens are co-infecting does not exclude the pathogenic effect of any of them. RVA and RVC are well-recognized intestinal pathogens for young piglets and what is debatable is the meaning of their detection in faecal samples. In this context, the investigation of the intensity of viral shedding (using qPCR approach) would probably provide additional and relevant information.
- - Lines 327-329: the sentence is not clear. “In our study, the detection of c. perfringens type A increased the probability of observing bacilli in close proximity to the mucosa decreased the probability of observing epithelial necrosis”
Author Response
Dear reviewer,
Many thanks for your precious comments which help us to improve our manuscript.
- The relevance of histological monitoring of lesions in the diagnosis of neonatal diarrhoea is demonstrated. Do the authors considered that quantitative detection using q-PCR would also provide relevant information? I think this point is interesting to discuss. Also, immunohistochemical detection of pathogens should be considered because it would be the gold standard method to evaluate the pathological role of different microorganisms. Can this method be applied in the histological samples for further research?
Of course probably that quantitative qPCR would provide relevant information, particularly for overgrowth of pathogens as previously described for Enterococcus hirae for example, but also of course for rotaviruses as you mention above.
Our data came from routine veterinary work in practice, unfortunately immunohistochemical examination is not routinely available in field laboratories (at least in France) but of course I hope that some research teams explore this method for example using RNA scope technology. In my knowledge, samples must be quite fresh to be used for RNA scope or such techniques so I could be an interesting prospective study for future researches.
- - The detection of etiological agents was limited to ETEC, C. perfringens, E. hirae, RVA and RVC. It is unclear why Clostridiodes difficile and enteropathogenic E. coli (EPEC) were not include in the research. The authors should clearly justify the selection of pathogens included in the research.
Culture with specific medium and toxins detection by PCR of Clostridioides difficile was not include in the research because C. difficile- like macro and microscopic lesions have not been observed routinely in French laboratories for a decade. We tried to precise this point in the discussion part of the revised manuscript.
EPEC are usually related with post-weaning diarrhoea as described in the Diseases of Swine, so it was not included in our research. But of course, we totally agree with you, EPEC isolates from neonatal diarrhoea cases have been already described by Wada et al., 2004, Alustiza et al., 2012, Mesoreno-Escuredo et al., 2018 and Vidal et al., 2019. We added this limitation in the discussion part of the revised manuscript.
- - In the discussion section (lines 383-389), it is mentioned the pathogenic role of C. difficile (not Clostridium but Clostridiodies) is debatable and unclear. However, it is generally considered that C. difficile is a relevant pathogen for young piglets (usually recognized as a more clearly pathogenic as compared to C. perfringens type A).
We agree with you of course, we tried to be more accurate in the revised manuscript.
Also, it is mentioned that C. difficile was not included in the research since there have been a significant reduction of antibiotic use in French pig herds. However, there is to my knowledge no relationship between the use of antibiotics and C. difficile detection in pigs has been demonstrated (in contrast to C. difficile infections in humans).
You are totally right. We delete this sentence to be more relevant.
- It is mentioned that “The role of RV as primary pathogens in neonatal diarrhoea is also reported as debatable” (lines 337-338). However, many reports have described their involvement in diarrhoea outbreaks in young animals, including both RVA and RVC. It is also mentioned that “The role of RVC in neonatal diarrhoea is controversial since, as in our study, Vidal et al found that RVC positive-cases were also positive for another enteric pathogen (RVA in their study)”. In my personal opinion, the fact that two different pathogens are co-infecting does not exclude the pathogenic effect of any of them. RVA and RVC are well-recognized intestinal pathogens for young piglets and what is debatable is the meaning of their detection in faecal samples. In this context, the investigation of the intensity of viral shedding (using qPCR approach) would probably provide additional and relevant information.
We agree with you, we change these sentences to be more precise. We hope this paragraph and the ideas are clearer now.
- Lines 327-329: the sentence is not clear. “In our study, the detection of c. perfringens type A increased the probability of observing bacilli in close proximity to the mucosa decreased the probability of observing epithelial necrosis”
A word is lacking. It’s corrected.

Reviewer 3 Report
The retrospective study presented by the authors described the aetiologies of neonatal diarrhoea cases and their associations with histological findings.
The manuscript is well written and presented in well-structured manner. The paper is easy to read. The tables and figures are clear presented. Conclusions are consistent with the presented results.
Specific comments:
1. Lacks detailed information about in-house multiplex PCRs for detection of E. coli and C. perfingens.
2. The abbreviations RVA and RVC should be explained.
Author Response
Dear Reviewer,
Many thanks for your time and your attentive review.
- I'm not sure to understand well what your are waiting for. You mean to precise the targets or something else. Both PCRs are not qPCRs so no Ct are available to characterize positivity. I can ask the two labs for this information if necessary.
- We precised the abbreviations at the end of the introduction part in the revised manuscript.
